# Experimental Evolution Reveals a Novel Ene Reductase That Detoxifies $\alpha,\beta$-Unsaturated Aldehydes in *Listeria monocytogenes*

Lei Sun,[a] Ann Van Loey,[a] Carolien Buvé,[a] Chris W. Michiels[a]

aDepartment of Microbial and Molecular Systems and Leuven Food Science and Nutrition Research Centre (LFoRCe), KU Leuven, Leuven, Belgium

**ABSTRACT** The plant essential oil component *trans*-cinnamaldehyde (t-CIN) exhibits antibacterial activity against a broad range of foodborne pathogenic bacteria, including *L. monocytogenes*, but its mode of action is not fully understood. In this study, several independent mutants of *L. monocytogenes* with increased t-CIN tolerance were obtained via experimental evolution. Whole-genome sequencing (WGS) analysis revealed single-nucleotide-variation mutations in the *yhfK* gene, encoding an oxidoreductase of the short-chain dehydrogenases/reductases superfamily, in each mutant. The deletion of *yhfK* conferred increased sensitivity to t-CIN and several other $\alpha,\beta$-unsaturated aldehydes, including *trans*-2-hexenal, citral, and 4-hydroxy-2-nonenal. The t-CIN tolerance of the deletion mutant was restored via genetic complementation with *yhfK*. Based on a gas chromatography-mass spectrometry (GC-MS) analysis of the culture supernatants, it is proposed that YhfK is an ene reductase that converts t-CIN to 3-phenylpropanal by reducing the C=C double bond of the $\alpha,\beta$-unsaturated aldehyde moiety. YhfK homologs are widely distributed in Bacteria, and the deletion of the corresponding homolog in *Bacillus subtilis* also caused increased sensitivity to t-CIN and *trans*-2-hexenal, suggesting that this protein may have a conserved function to protect bacteria against toxic $\alpha,\beta$-unsaturated aldehydes in their environments.

**IMPORTANCE** While bacterial resistance against clinically used antibiotics has been well studied, less is known about resistance against other antimicrobials, such as natural compounds that could replace traditional food preservatives. In this work, we report that the food pathogen *Listeria monocytogenes* can rapidly develop an elevated tolerance against t-cinnamaldehyde, a natural antimicrobial from cinnamon, by single base pair changes in the yhfK gene. The enzyme encoded by this gene is an oxidoreductase, but its substrates and precise role were hitherto unknown. We demonstrate that the enzyme reduces the double bond in t-cinnamaldehyde and thereby abolishes its antibacterial activity. Furthermore, the mutations linked to t-CIN tolerance increased bacterial sensitivity to a related compound, suggesting that they modify the substrate specificity of the enzyme. Since the family of oxidoreductases to which YhfK belongs is of great interest in the mediation of stereospecific reactions in biocatalysis, our work may also have unanticipated application potential in this field.

**KEYWORDS** *Listeria monocytogenes*, natural antimicrobials, food preservatives, antimicrobial resistance, ene reductase, short-chain dehydrogenase/reductase, SDR

Address correspondence to Chris W. Michiels, chris.michiels@kuleuven.be.

The authors declare no conflict of interest.

*L*isteria monocytogenes is a Gram-positive foodborne pathogen that causes severe, invasive listeriosis and meningitis among susceptible persons, such as immuno-compromised individuals, pregnant women, and elderly persons (1, 2). The organism thrives well in a wide range of natural environments, including soil, freshwater, decaying plant material, and the gastrointestinal tracts of various animals. It also thrives as a

member of the resident house microbiota of food production facilities. Therefore, it is therefore a common contaminant during food production and storage processes (2, 3). Furthermore, the ability of *L. monocytogenes* to grow under adverse conditions, such as high salt concentration (up to 10% NaCl) and low temperature (as low as 0°C) make this pathogen a major concern in refrigerated, ready-to-eat foods. Preservatives, such as nitrites, benzoates, and sorbates are commonly used to prevent its outgrowth to high numbers, but these compounds are increasingly under scrutiny for possible adverse health effects, and food producers are exploring more natural alternatives with which to replace them (4, 5).

Plant essential oil constituents have received much attention in this respect, and a well-studied example is *trans*-cinnamaldehyde (t-CIN) ([E]-3-phenyl-2-propenal), which is one of the major components of cinnamon essential oil (4, 6). Previous work in our laboratory showed that the growth inhibition by t-CIN against *L. monocytogenes* is typically characterized by a dose-dependent elongation of the lag phase and a reduction of the growth rate (7). Many different effects of t-CIN on bacterial cells have been reported, including an increased cell membrane permeability (8), the inhibition of membrane-associated ATPase activity (9), elevated intracellular redox stress (10, 11), the inhibition of cell division (12), the repression of quorum sensing systems (13, 14), and the disruption of cell wall homeostasis (15). However, these are quite broad general defects that are induced by many different types of antimicrobials and are therefore likely to be secondary effects that do not reflect the specific mode of action or the primary cellular targets of t-CIN (16). Besides t-CIN, other $\alpha,\beta$-unsaturated carbonyl compounds of plant origin, such as *trans*-2-hexenal (t-HEX) and citral, exhibit broad antimicrobial activity that offers promise for applications (4, 17). The $\alpha,\beta$-unsaturated carbonyl moiety shared by these compounds is electrophilic and accounts for reactivity with a wide range of nucleophilic nitrogen and sulfur atoms in biomolecules, including proteins, glutathione, and cysteine, primarily by the Michael-type addition reaction (18). Proteins containing active Cys residues (e.g., glyceraldehyde-3-phosphate dehydrogenase, thioredoxins, and glutaredoxins) have particularly been identified as intracellular targets of various electrophilic $\alpha$, $\beta$-unsaturated carbonyls in human and animal cells, but the targets in bacteria have not been elucidated (19).

Apart from conjugation with cellular nucleophiles, $\alpha,\beta$-unsaturated aldehydes can be detoxified via enzymatic transformation to less electrophilic molecules in at least some microbes. It has been demonstrated that t-CIN can be reduced to the less toxic 3-phenyl-2-propenol by *Escherichia coli* O157:H7 (20) and by the fungi *Aspergillus ochraceus* and *Penicillium expansum* (21, 22). A predicted NADPH-dependent aldehyde reductase, YqhD, was anticipated to catalyze this reaction in *E. coli* because the transcription of *yqhD* was significantly induced by t-CIN, and the inactivation of the gene significantly reduced the tolerance of *E. coli* to t-CIN (11, 20). Moreover, a broad range of short-chain aldehydes, including acrolein, isobutyraldehyde, glycolaldehyde, butanaldehyde, malondialdehyde, and propanaldehyde, were previously demonstrated to be reduced by purified YqhD from *E. coli in vitro* (23, 24). Besides, the Old Yellow Enzyme family of flavin-dependent NADPH dehydrogenases, which are widely distributed in bacteria and fungi, is comprised of several ene reductases that mediate the hydrogenation of the C=C double bond of a wide spectrum of $\alpha,\beta$-unsaturated aldehydes in the presence of cofactors (25, 26). Examples include YqjM from *B. subtilis* (27), XenA from *Pseudomonas putida*, KYE1 from *Kluyveromyces lactis*, and Yers-ER from *Yersinia bercovieri* (28), all of which exhibit versatile substrate specificity toward $\alpha,\beta$-unsaturated carbonyl compounds including t-CIN, as shown via *in vitro* enzymatic assays. However, so far, no enzymes that are able to transform $\alpha,\beta$-unsaturated carbonyl compounds have been reported in *L. monocytogenes*.

In an attempt to generate novel insights into the antibacterial mechanisms of t-CIN or in the bacterial defense against the compound, *L. monocytogenes* Scott A was subjected to experimental evolution to develop increased tolerance to t-CIN, and the mutations that were responsible for this increased tolerance were identified in this

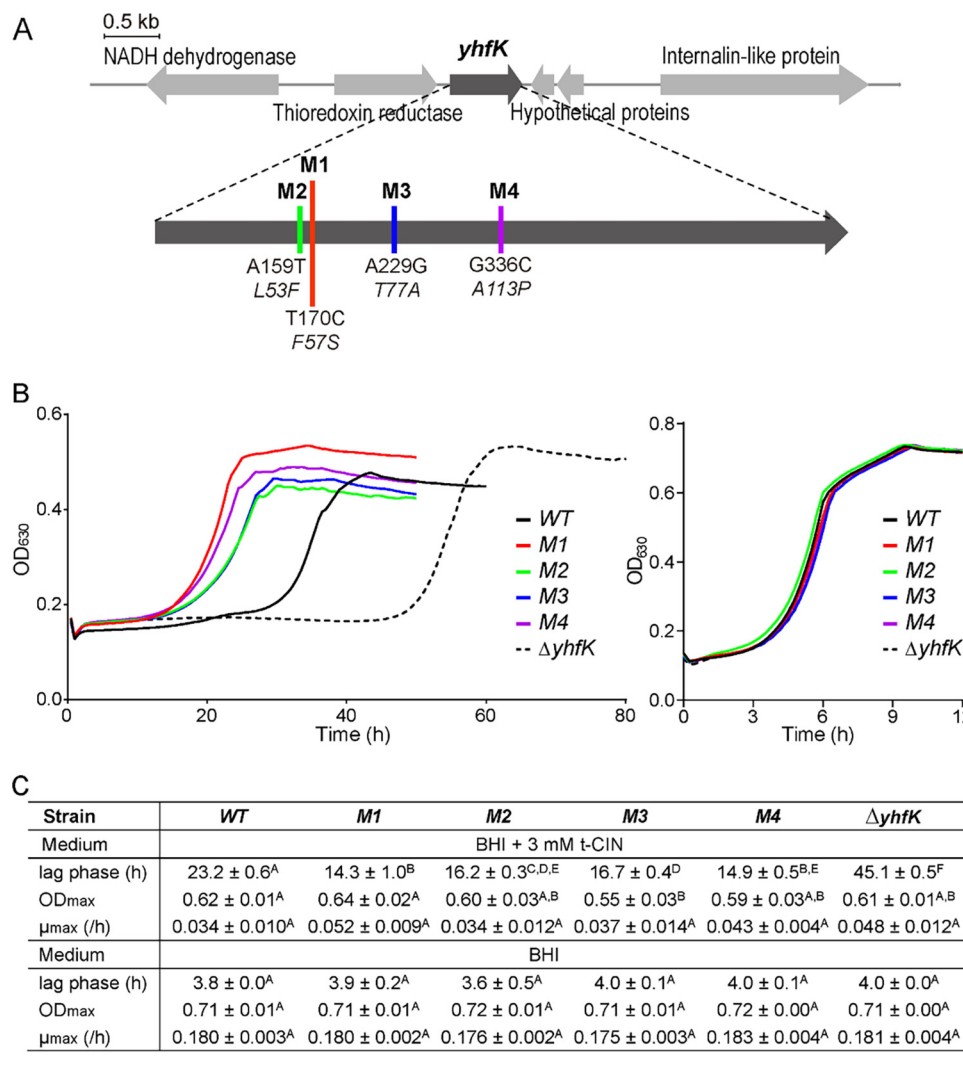

**FIG 1** *L. monocytogenes* Scott A mutants with increased t-CIN resistance have mutations in *yhfK* that have resulted in single amino acid changes. (A) *yhfK* gene context and mutations in each of four selected mutants (M1, M2, M3, M4). Colored vertical lines show the positions of the mutations, with specific base changes (straight font type) and corresponding amino acid changes (italic) specified under each line. (B) Growth curves of the WT strain and t-CIN resistant mutants in BHI with (left) and without (right) 3 mM t-CIN. (C) Growth parameters ($\lambda$, $\mu$max, and $OD_{max}$), represented as the mean $\pm$ SD ($n = 3$). Values followed by a common letter are not significantly different at the 5% level.

work. This led to the identification of an ene reductase of the short-chain dehydrogenases/reductase (SDR) superfamily that reduces t-CIN and several other $\alpha,\beta$-unsaturated aldehydes, thereby protecting *L. monocytogenes* against their toxic activity.

## RESULTS

**Evolution of *L. monocytogenes* for increased t-CIN tolerance selects for YhfK variant proteins.** After being subcultured for nine rounds in 3 mM t-CIN or for seven rounds in 4 mM t-CIN, mutants with significantly elevated t-CIN tolerance appeared in seven independent lineages (Fig. 1). This was not the case in the control lineages without t-CIN. Four evolved mutants were selected and further characterized. Their lag phase ($\lambda$) in BHI with 3 mM t-CIN was reduced to 62 to 72% of that of the wild-type (WT) strain, whereas their exponential growth rates ($\mu$max) and maximal optical density ($OD_{max}$) values were unaffected, except for a slightly lower $OD_{max}$ for mutant M3 (Fig. 1). All the mutants showed WT growth in the absence of t-CIN. Interestingly, a WGS analysis revealed that the mutants had each acquired a single base change in a different position in the coding region of the *yhfK* gene (NCBI locus CRH05_RS12830),

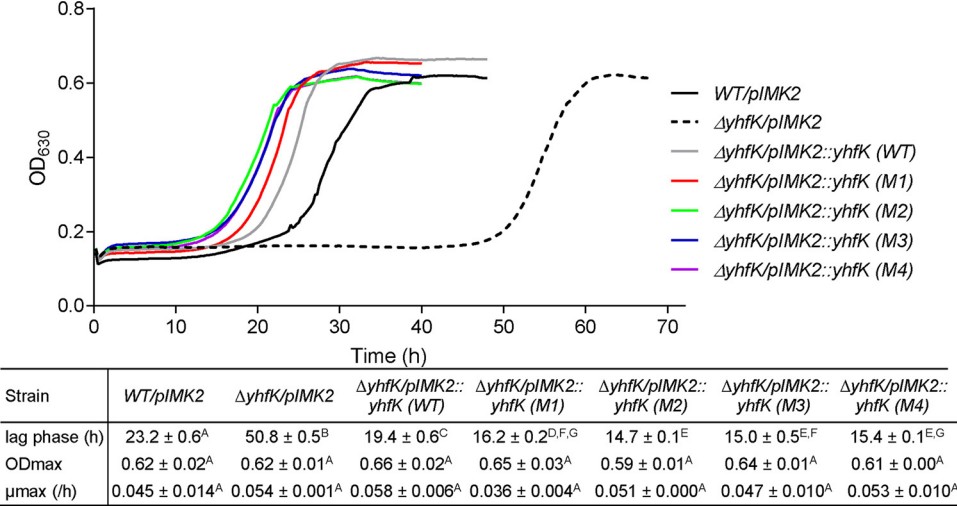

| Strain | WT/pIMK2 | ΔyhfK/pIMK2 | ΔyhfK/pIMK2::yhfK (WT) | ΔyhfK/pIMK2::yhfK (M1) | ΔyhfK/pIMK2::yhfK (M2) | ΔyhfK/pIMK2::yhfK (M3) | ΔyhfK/pIMK2::yhfK (M4) |
|---|---|---|---|---|---|---|---|
| lag phase (h) | $23.2 \pm 0.6^A$ | $50.8 \pm 0.5^B$ | $19.4 \pm 0.6^C$ | $16.2 \pm 0.2^{D,F,G}$ | $14.7 \pm 0.1^E$ | $15.0 \pm 0.5^{E,F}$ | $15.4 \pm 0.1^{E,G}$ |
| ODmax | $0.62 \pm 0.02^A$ | $0.62 \pm 0.01^A$ | $0.66 \pm 0.02^A$ | $0.65 \pm 0.03^A$ | $0.59 \pm 0.01^A$ | $0.64 \pm 0.01^A$ | $0.61 \pm 0.00^A$ |
| μmax (/h) | $0.045 \pm 0.014^A$ | $0.054 \pm 0.001^A$ | $0.058 \pm 0.006^A$ | $0.036 \pm 0.004^A$ | $0.051 \pm 0.000^A$ | $0.047 \pm 0.010^A$ | $0.053 \pm 0.010^A$ |

**FIG 2** Complementation analysis of *L. monocytogenes* Δ*yhfK* with WT or mutant *yhfK* alleles, based on growth curves recorded at 30°C in BHI with 3 mM t-CIN. The control strains had an empty pIMK2 plasmid. For clarity, the standard deviations are not shown in the graphs. The growth parameters ($\lambda$, $\mu_{max}$, and $OD_{max}$) are represented in the table as the mean $\pm$ SD ($n = 3$). Values followed by a common letter are not significantly different at the 5% level.

resulting in four single amino acid variants of YhfK (Fig. 1). YhfK is predicted to be an oxidoreductase of the large, diverse SDR enzyme family.

Further evidence implicating YhfK in t-CIN tolerance was obtained when *yhfK* was deleted, as this resulted in an almost doubling of the lag phase (50 h for Δ*yhfK* versus 28 h for the WT strain) in the presence of 3 mM t-CIN, whereas no growth attenuation was noted in the absence of t-CIN (Fig. 2). The complementation of the Δ*yhfK* strain with the WT *yhfK* gene (strain Δ*yhfK/pIMK2::yhfK*) again reduced the lag phase, making it 3 h shorter than that of the WT, possibly because the gene was under the control of a strong constitutive promoter in pIMK2 (29). The complementation of Δ*yhfK* with each of the four mutant *yhfK* alleles reduced the lag times even further, by 7 to 8 h, relative to the WT. This was in line with their ability to confer t-CIN tolerance in the evolved mutants. Complementation with neither of the *yhfK* alleles affected the growth of the Δ*yhfK* strain in BHI (Fig. S1). However, little is known about the function of YhfK or its homologs in bacteria, except that it is part of the sigma B regulon in *L. monocytogenes* (30–32). In addition, there are studies indicating that the gene is induced in acidic conditions and that its deletion confers moderate sensitivity of *L. monocytogenes* to severe acid shock (pH 2.5) (30, 33). Here, we compared the growth of the WT and Δ*yhfK* strains in BHI acidified with HCl to a pH of 4.0, but no difference was observed (Fig. S2), suggesting that YhfK only plays a role in lethal acid shocks.

**YhfK provides tolerance to multiple $\alpha,\beta$-unsaturated aldehydes but not to other thiol-reactive electrophiles.** The evolved t-CIN resistant strains as well as the Δ*yhfK* deletion strain and its various complemented derivatives were used to assess the role of YhfK in tolerance of *L. monocytogenes* to $\alpha,\beta$-unsaturated aldehydes other than t-CIN as well as to thiol-reactive compounds with other electrophilic functional groups (Fig. 3, Fig. 4; Fig. S3). 6 mM t-HEX induced a considerable lag phase extension in the Δ*yhfK* strain (42 h), relative to the WT (15 h) and the complemented strain Δ*yhfK/pIMK2::yhfK* (WT) (16 h) (Fig. 3B). Surprisingly, the evolved t-CIN tolerant strains were more sensitive to t-HEX than was the WT strain, with the lag times being lengthened by 10 to 18 h (Fig. 3A). We speculate that the opposite effect of the amino acid substitutions on the tolerances to t-CIN and t-HEX may reflect an altered YhfK substrate specificity of the enzyme. The Δ*yhfK* strain also showed increased sensitivity, again reflected by an extended lag phase, to citral (Fig. S3). A fourth $\alpha,\beta$-unsaturated aldehyde that was tested is 4-HNE, which is a degradation product of polyunsaturated fatty acids that is formed during the oxidative burst as part of the cellular immunity (34, 35).

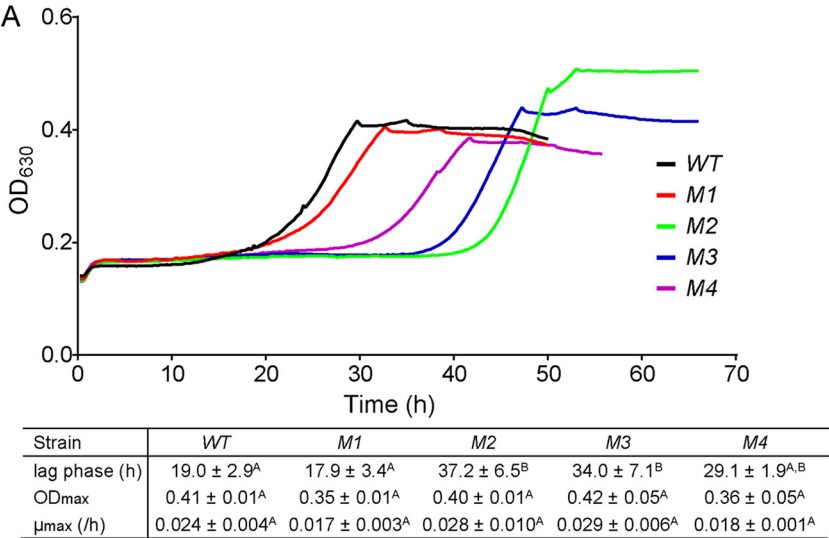

| Strain | WT | M1 | M2 | M3 | M4 |
|---|---|---|---|---|---|
| lag phase (h) | $19.0 \pm 2.9^A$ | $17.9 \pm 3.4^A$ | $37.2 \pm 6.5^B$ | $34.0 \pm 7.1^B$ | $29.1 \pm 1.9^{A,B}$ |
| ODmax | $0.41 \pm 0.01^A$ | $0.35 \pm 0.01^A$ | $0.40 \pm 0.01^A$ | $0.42 \pm 0.05^A$ | $0.36 \pm 0.05^A$ |
| $\mu$max (/h) | $0.024 \pm 0.004^A$ | $0.017 \pm 0.003^A$ | $0.028 \pm 0.010^A$ | $0.029 \pm 0.006^A$ | $0.018 \pm 0.001^A$ |

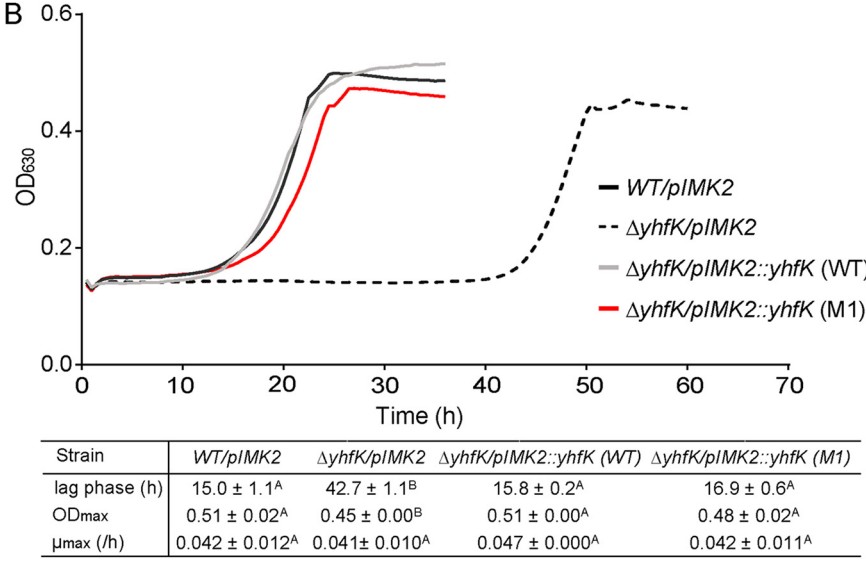

| Strain | WT/pIMK2 | $\Delta yhfK$/pIMK2 | $\Delta yhfK$/pIMK2::yhfK (WT) | $\Delta yhfK$/pIMK2::yhfK (M1) |
|---|---|---|---|---|
| lag phase (h) | $15.0 \pm 1.1^A$ | $42.7 \pm 1.1^B$ | $15.8 \pm 0.2^A$ | $16.9 \pm 0.6^A$ |
| ODmax | $0.51 \pm 0.02^A$ | $0.45 \pm 0.00^B$ | $0.51 \pm 0.00^A$ | $0.48 \pm 0.02^A$ |
| $\mu$max (/h) | $0.042 \pm 0.012^A$ | $0.041 \pm 0.010^A$ | $0.047 \pm 0.000^A$ | $0.042 \pm 0.011^A$ |

**FIG 3** Growth curves showing the involvement of YhfK in the t-HEX tolerance of *L. monocytogenes*. (A) Comparison of the WT strain with different t-CIN-tolerant evolved mutants. (B) Comparison of WT/pIMK2, $\Delta yhfK$/pIMK2, and $\Delta yhfK$ complemented with different variants of the YhfK protein. Cultures were grown in BHI broth with 6 mM t-HEX at 30°C. The curve represents the mean value of measurements of three independent cultures. The growth parameters ($\lambda$, $\mu$max, and ODmax) are represented as the mean $\pm$ SD ($n = 3$). Values followed by a common letter are not significantly different at the 5% level.

Since the compound is bactericidal at low concentrations and may play a role in pathogen elimination during infection, we tested its activity in a killing assay rather than in a growth inhibition assay (Fig. 4). Upon incubation with 2 mM 4-HNE for 6 h, survival was only 25% for the $\Delta yhfK$ strain, compared to 40% and 53% for the WT and the complemented strains, respectively. Finally, allyl isothiocyanate and diamide, two antimicrobials that are also thiol-reactive but that lack an $\alpha,\beta$-unsaturated aldehyde moiety, were also tested, but they did not have a different activity against the $\Delta yhfK$ strain or the the WT strain (Fig. S3). Overall, the above findings indicate a role of YhfK in the tolerance of *L. monocytogenes* to antimicrobial $\alpha,\beta$-unsaturated aldehydes.

**YhfK reduces the C=C double bond in t-CIN.** Since YhfK is a predicted oxidoreductase of the SDR family, and given its role in the tolerance to $\alpha,\beta$-unsaturated aldehydes, we speculated that it could possibly catalyze the reduction of either the ene or the carbonyl group of t-CIN. To investigate this, the degradation of t-CIN and the formation of the three possible reduction products were analyzed using GC-MS in cultures of *WT/*

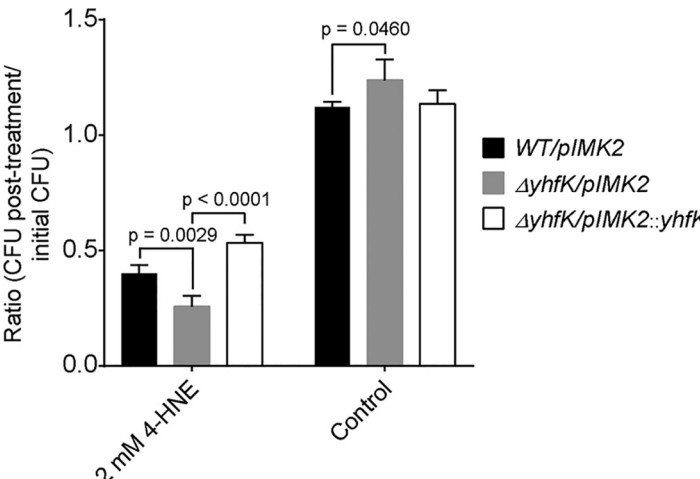

**FIG 4** Survival assay of *WT/pIMK2*, Δ*yhfK/pIMK2*, and Δ*yhfK/pIMK2::yhfK* treated with 2 mM 4-HNE at 30°C for 6 h. The *y* axis represents the fraction of survivors after 6 h. The data are represented as the mean ± SD (*n* = 3). The *P* values from a two-tailed Student's *t* test are indicated.

*pIMK2*, Δ*yhfK/pIMK2*, and Δ*yhfK/pIMK2::yhfK* strains that were grown with 1 mM t-CIN (Fig. 5). The analysis of a sterile medium control indicated that the concentration of t-CIN declined to about 0.7 mM in the first 6 h and then stayed more or less constant until the end of the measurements (24 h). Since the same initial decline occurred in all of the cultures, it is probably the result of reactions of t-CIN with nucleophiles in the BHI broth. Similar losses of t-CIN during culture preparation and the first hours of incubation have previously been reported (20). In the culture of the WT strain (*WT/pIMK2*), the t-CIN concentration started to drop further at the end of the exponential growth phase (18 h), reaching 0.28 mM in the early stationary phase (24 h). Importantly, the three t-CIN reduction products appeared in the culture, suggesting that *L. monocytogenes* can reduce both the aldehyde and the ene group of t-CIN. It should be noted that the reduction products are reported as relative concentrations because we did not make standard curves for these compounds. Therefore, a quantitative analysis of the formation of the reduction products is not possible. The Δ*yhfK/pIMK2* strain also degraded t-CIN during the transition from the exponential phase to the stationary phase, but it did so slower than did *WT/pIMK2*, resulting in a final concentration of 0.43 mM after 24 h. However, the most remarkable difference with the WT strain was that the deletion strain only produced 3-phenyl-2-propenol but not the two other reduction products, indicating that it could still reduce the aldehyde group but not the ene group. The complemented deletion strain (Δ*yhfK/pIMK2::yhfK*) finally, degraded t-CIN earlier and more rapidly than did both of the other strains, and it produced the three reduction products, albeit in different relative amounts than did the WT. 3-phenylpropanal was already detected at 6 h and peaked at 12 h at a much higher level than was observed in the WT culture. Later, at 24 h, t-CIN declined to an almost undetectable level at, while 3-phenylpropanol started to appear and reached a maximal level at 24 h. 3-phenyl-2-propenol was also formed, but this was at clearly lower levels than those that were observed in the WT strain. The higher production of 3-phenylpropanal and lower production of 3-phenyl-2-propenol in the culture of the complemented strain, compared to the WT, can be explained by the strong expression of YhfK by the pIMK2 promoter. These data show that *L. monocytogenes* can metabolize t-CIN by a combination of ene and aldehyde reduction, without a strict order for both reactions. They also indicate that YhfK is an ene reductase that reduces the C=C double bond in t-CIN (Fig. 6).

The antimicrobial activity of the three t-CIN metabolites was also tested. Neither alcohol showed any activity up to 50 mM (data not shown), whereas 3-phenylpropanal retained some activity but was less potent than was t-CIN (Fig. S4, and comparison with Fig. 1 for t-CIN). Furthermore, the antimicrobial activity of 3-phenylpropanal was not affected by the deletion or the overexpression of YhfK. As a result, the sensitivity of

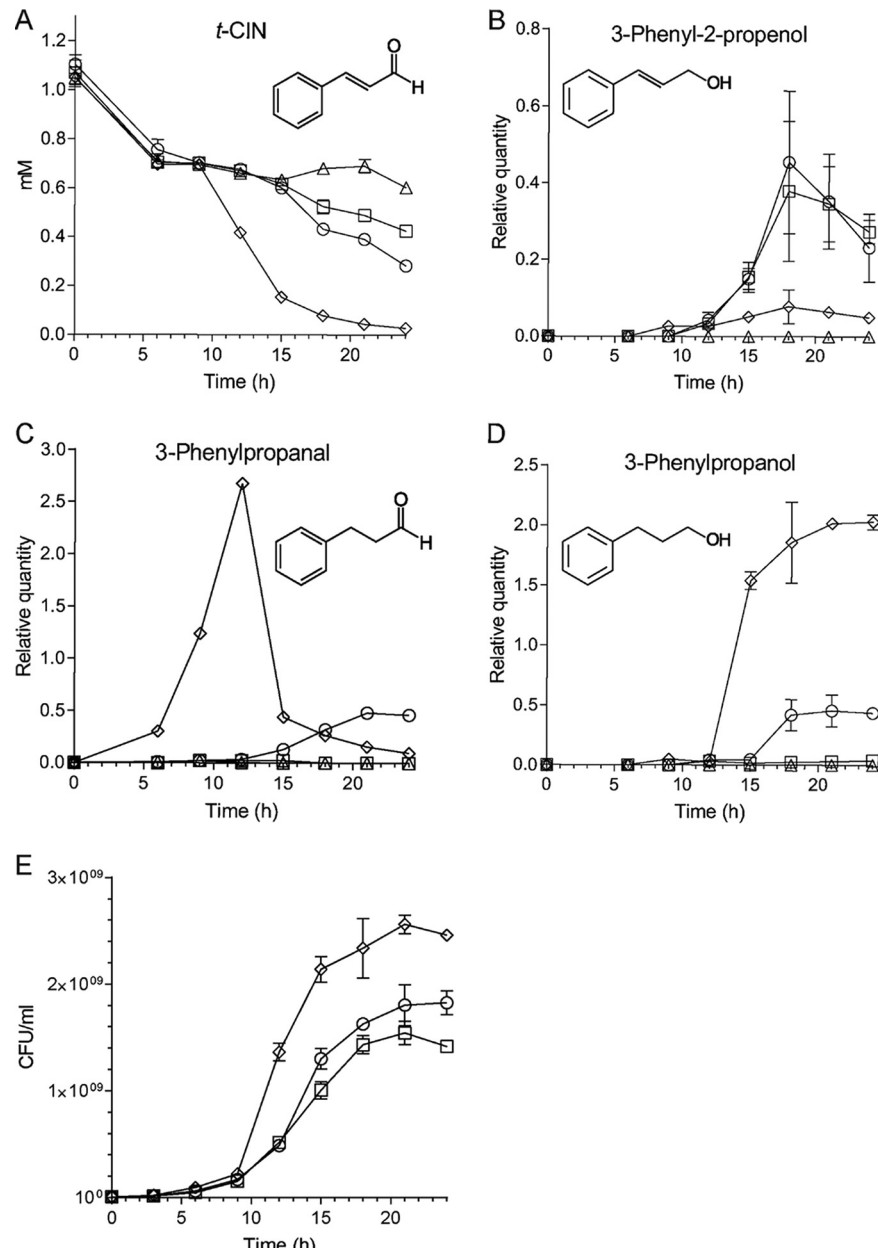

**FIG 5** GC-MS analysis of t-CIN degradation in cultures of *L. monocytogenes WT/pIMK2*, *ΔyhfK/pIMK2*, and *ΔyhfK/pIMK2::yhfK* grown in BHI with 1 mM t-CIN at 30°C. Sterile BHI medium with 1 mM t-CIN was included as the control. (A) Concentration of t-CIN. (B) Relative concentration of 3-phenyl-2-propenol. (C) Relative concentration of 3-phenylpropanal. (D) Relative concentration of 3-phenylpropanol. (E) CFU enumeration. For panels B to D, the *y* axis represents the ratio of the peak area of the corresponding t-CIN degradation product in the chromatogram to the peak area of the ethyl benzoate internal standard and should therefore be regarded as a relative concentration. The lines connecting the data points were added for visual guidance only. All data represent the mean value of three independent cultures ± SD.

the strains for t-CIN (as reflected by their growth curves in Fig. 5E) correlated with their capacities to metabolize t-CIN. The fastest and most complete conversion of t-CIN was by the complemented strain, and this strain also showed the fastest growth in the presence of t-CIN. Conversely, the *yhfK* deletion strain had the slowest metabolism of t-CIN and also showed the slowest growth with t-CIN. The t-CIN metabolism and growth with t-CIN of the WT strain were intermediate. Taken together, these data suggest that a possible biological function of YhfK in *L. monocytogenes* is to detoxify t-CIN and other $\alpha,\beta$-unsaturated aldehydes in the environment.

**FIG 6** Proposed scheme of t-CIN metabolism in *L. monocytogenes*. YhfK is suggested to be an ene reductase that converts t-CIN to 3-phenylpropanal. Since both 3-phenyl-2-propenol and 3-phenylpropanol are produced by WT *L. monocytogenes*, the organism must also have one or more aldehyde reductases.

**A YhfK homolog in *B. subtilis* also confers tolerance to t-CIN and t-HEX.** *B. subtilis* has a homolog that shares 43% amino acid sequence identity with YhfK of *L. monocytogenes*, and, like in *L. monocytogenes*, the deletion of the *yhfK* gene strongly increased the sensitivity of *B. subtilis* to t-CIN and t-HEX (Fig. 7), suggesting that YhfK has a conserved function in both bacteria. Attenuated growth was exhibited by both mutant strains when incubating with t-CIN and t-HEX, in comparison with the parental strain. In the presence of 3 mM t-CIN and 2 mM t-HEX, the *B. subtilis* WT strain showed an extended lag phase and a reduced exponential growth rate, whereas both Δ*yhfK* strains failed to grow for at least up to 60 h. Similar to what was observed in *L. monocytogenes*, the mutants showed no growth deficiency in the absence of t-CIN and t-HEX. These results suggest a conserved function for YhfK in bacteria.

## DISCUSSION

Like many other plant essential oil compounds, t-CIN and t-HEX exhibit antimicrobial activity toward *L. monocytogenes* and other foodborne pathogens; therefore, they may be applied in natural food preservative systems (4, 6). In the present study, increased tolerance toward t-CIN was evolved in *L. monocytogenes* and shown to be associated with single nucleotide variants of *yhfK*, which is a gene coding for a putative oxidoreductase of the SDR family with unknown function. The deletion of *yhfK* did not affect the normal growth of *L. monocytogenes*, but it did cause increased sensitivity to $\alpha,\beta$-unsaturated aldehyde compounds, including t-CIN, t-HEX, citral, and 4-HNE. A GC-MS analysis of the culture supernatants indicated that YhfK mediates the reduction of t-CIN to 3-phenylpropanal, which is then further converted by *L. monocytogenes* to 3-phenylpronanol, thereby indicating that YhfK is a NAD(P)H-dependent ene reductase.

Interestingly, low concentrations of 3-phenyl-2-propenol were also detected in the culture supernatants of *L. monocytogenes* that were grown with t-CIN (Fig. 5B). This compound was, in fact, a major degradation product when *E. coli* O157:H7 was incubated with t-CIN (20). Although 3-phenylpropanal was not analyzed in the latter study, it can be deduced from the presented data that 3-phenylpropanal was either absent or produced in small amounts (20). Correspondingly, we could not find a YhfK homolog (with identity ≥ 30%) in *E. coli* O157:H7 (data not shown). Two dehydrogenase/reductase enzymes, namely, YqhD and DkgA, were postulated to possibly convert t-CIN to 3-phenyl-2-propenol in *E. coli*, as a transcriptional analysis showed the corresponding genes to be induced by t-CIN (11, 20). Furthermore, the inactivation of *yqhD* significantly increased the sensitivity of *E. coli* to t-CIN (11). The purified YqhD protein of *E. coli* had previously been found to catalyze the reduction of a broad range of short-chain aldehydes, including acrolein, isobutyraldehyde, glycolaldehyde, butanaldehyde, malondialdehyde, and propanaldehyde, to the corresponding alcohols in a NADPH-dependent manner (23, 24). DkgA, on the other hand, is an aldo-keto reductase family enzyme that uses NADPH as its preferred cofactor and shows activity toward a variety

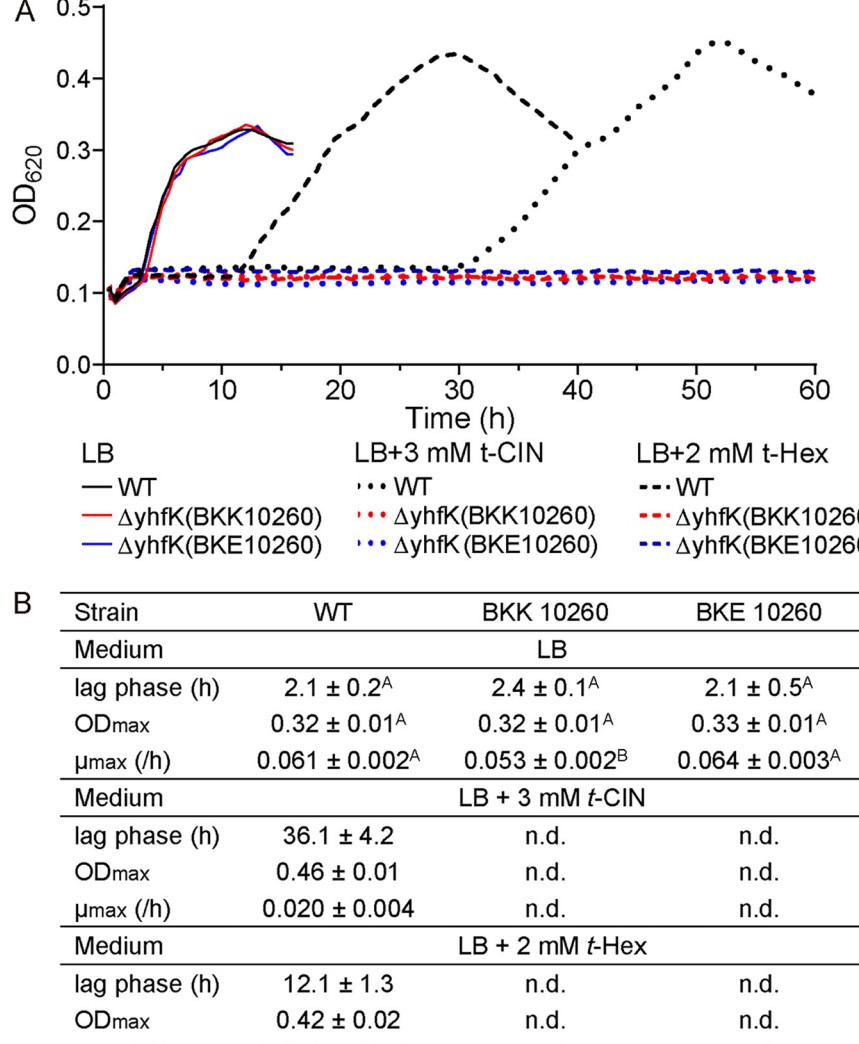

| Strain | WT | BKK 10260 | BKE 10260 |
|---|---|---|---|
| Medium | LB | | |
| lag phase (h) | 2.1 ± 0.2[A] | 2.4 ± 0.1[A] | 2.1 ± 0.5[A] |
| OD$_{max}$ | 0.32 ± 0.01[A] | 0.32 ± 0.01[A] | 0.33 ± 0.01[A] |
| µ$_{max}$ (/h) | 0.061 ± 0.002[A] | 0.053 ± 0.002[B] | 0.064 ± 0.003[A] |
| Medium | LB + 3 mM *t*-CIN | | |
| lag phase (h) | 36.1 ± 4.2 | n.d. | n.d. |
| OD$_{max}$ | 0.46 ± 0.01 | n.d. | n.d. |
| µ$_{max}$ (/h) | 0.020 ± 0.004 | n.d. | n.d. |
| Medium | LB + 2 mM *t*-Hex | | |
| lag phase (h) | 12.1 ± 1.3 | n.d. | n.d. |
| OD$_{max}$ | 0.42 ± 0.02 | n.d. | n.d. |
| µ$_{max}$ (/h) | 0.023 ± 0.002 | n.d. | n.d. |

**FIG 7** (A) Growth curves of *B. subtilis* 168 and its mutants ΔyhfK::Km and ΔyhfK::Erm in the absence and presence of t-CIN and t-HEX at 37°C. Each curve represents a single culture but is representative of three independent cultures. (B) Growth parameters ($\lambda$, $\mu_{max}$, and OD$_{max}$) were calculated from the growth curves and are represented as the mean ± SD; $n = 3$. Values followed by a common letter are not significantly different at the 5% level.

of aldehydes and ketones, such as methylglyoxal (36). However, direct evidence showing that either YqhD or DkgA is active on t-CIN or other $\alpha,\beta$-unsaturated aldehydes is still lacking. A close homolog of *E. coli* O157:H7 YqhD, which shares a 57.47% identity (coverage, 99%; E value, 2E$^{-153}$) exists in *L. monocytogenes* Scott A (NCBI accession: EGJ25272). As for the DkgA of *E. coli* O157:H7, four homologs (NCBI accession: EGJ26117.1 [E value: 3E$^{-68}$], EGJ25830.1 [E value: 7E$^{-72}$], EGJ26226.1 [E value: 1E$^{-80}$], and EGJ24325.1 [E value: 2E$^{-90}$]) exist in *L. monocytogenes* Scott A with identities ranging from 40.61% to 45.39% (coverage ≥ 93%). However, none of these homologs in *L. monocytogenes* have been characterized, and mutations in the corresponding genes were not retrieved in our evolution experiments. So, it remains unclear whether they play a role in the detoxification of t-CIN or other aldehydes. Several fungal species, including *A. ochraceus* (21) and *P. expansum* (22), were reported to transform t-CIN to 3-phenyl-2-propenol, as well, presumably by means of aldehyde reductase enzymes, suggesting that the conversion to 3-phenyl-2-propenol might be a common detoxification mechanism of t-CIN in many microbes. Versatile aldehyde reductase enzymes are

widely distributed in virtually all organisms, and, since some of them have a relaxed substrate selectivity, they can be anticipated to detoxify exogenous aldehydes.

The deletion of *yhfK* also resulted in increased sensitivity to 4-HNE in *L. monocytogenes* (Fig. 4). 4-HNE is commonly generated during the oxidative deterioration of polyunsaturated fatty acids, together with some other $\alpha,\beta$-unsaturated aldehydes, including acrolein and 4-hydroxy-2-hexenal (34). The $\alpha,\beta$-unsaturated aldehyde group of 4-HNE easily undergoes a Michael-type addition reaction with thiol and amino groups of proteins to generate protein adducts, and this reactivity is held responsible for its antibacterial activity (34, 37, 38). A recent study found that two NADPH-dependent oxidoreductases, namely, the flavin-dependent enone reductase Rha1 and the alcohol/quinone reductase Rha2, play pivotal roles in the resistance of *L. monocytogenes* against 4-HNE (35). Recombinant Rha1 and Rha2 reduced the C=C double bond of 4-HNE, generating 4-hydroxynonanal in a NADPH-dependent manner. However, against 4-hydroxy-2-hexenal, Rha1 and Rha2 exhibited no activity and modest activity, respectively (35), indicating the narrow substrate specificities of the two enzymes. The expression of both Rha1 and Rha2 was highly induced (238-fold and 180-fold, respectively) in response to 640 $\mu$M 4-HNE, as revealed via a transcriptome analysis. Interestingly, *yhfK* was also found to be induced 19-fold in the same study, but its involvement in 4-HNE degradation was not investigated or suggested. Since our work shows that the absence of YhfK completely abolished the production of 3-phenylpropanal from t-CIN, it can be concluded that none of Rha1, Rha2, or any other ene reductases reduce the double bond of t-CIN in *L. monocytogenes*. Although a role in virulence could not be demonstrated, Rha1 and Rha2 were shown to contribute to the tolerance of *L. monocytogenes* to 4-HNE *in vitro*, and the heterologous expression of the genes promoted the survival of *B. subtilis*, following phagocytosis by macrophages (35). Since our present work shows that YhfK also protects *L. monocytogenes* against 4-HNE, it would be worthwhile to investigate whether this enzyme also contributes to pathogenicity.

Microbial ene reductases acting on $\alpha,\beta$-unsaturated carbonyl compounds commonly belong to the flavin-dependent OYEs, which are renowned for containing a noncovalently bound FMN cofactor (25, 26). These flavoenzymes catalyze the *trans*-hydrogenation of the C=C double bond of $\alpha,\beta$-unsaturated carbonyl compounds through the reductive half-reaction, using NAD(P)H as the reductant of FMN (26, 39). YqjM of *B. subtilis*, one of the most thoroughly studied OYEs, exhibits activities toward an array of $\alpha,\beta$-unsaturated aldehydes and ketones, including t-CIN, t-HEX, and cyclohexenone (27, 40, 41). An additional OYE from *B. subtilis* (YqiG) also showed considerable activity toward alpha-methyl cinnamaldehyde, citral, and cyclohexenone (42). Unlike in *B. subtilis*, OYEs have not been reported in *L. monocytogenes*, as far as we know, but a BLASTP analysis identified two homologs of YqjM and YqiG in *L. monocytogenes* EGD-e, namely, lmo2471 (NCBI accession, WP_003732493; identity, 63% and 30% for YqjM and YqiG, respectively) and lmo0489 (NCBI accession, WP_010989492; identity, 35% and 33% for YqjM and YqiG, respectively). The reactions that are catalyzed by these enzymes as well as their substrate specificity remain to be elucidated.

Four independent mutants with increased t-CIN tolerance that was obtained via experimental evolution each contained a base change in *yhfK* that resulted in a single amino acid substitution (L53F, F57S, T77A, A113P) (Fig. 1). Since we demonstrated that YhfK is an ene reductase that detoxifies t-CIN, it seems likely that the increased t-CIN tolerance of the evolved mutants stems from a higher t-CIN conversion rate by the variant YhfK proteins. Remarkably, the tolerance of the mutant strains toward t-HEX was unaltered (YhfK$_{F57S}$) or reduced (YhfK$_{L53F}$, YhfK$_{T77A}$, and YhfK$_{A113P}$) (Fig. 3), suggesting unaltered or reduced t-HEX conversion rates by the corresponding YhfK variants. To understand how the amino acid substitutions affect the structure and catalytic function of YhfK, crystal structures of YhfK in complex with and without cofactors and substrates are needed. Nevertheless, even without this detailed insight, our results imply the feasibility of improving the enzymatic activity of YhfK toward specific substrates via natural evolution or rational mutagenesis. The increased sensitivity of a $\Delta yhfK$ mutant not only to t-CIN but also to t-HEX, citral, and 4-HNE indicates that YhfK has a

relatively relaxed substrate specificity. Because of their abilities to stereoselectively reduce activated C=C double bonds, several ene reductases, particularly OYEs, are interesting biocatalysts to produce high-value specialty chemicals (26, 43–45). YhfK may represent an interesting new scaffold of the SDR superfamily for the development of novel biocatalysts for the reduction of $\alpha,\beta$-unsaturated carbonyl compounds.

The results of a BLASTP analysis show that YhfK-like enzymes are widely distributed in many phyla of Bacteria and even exist in some Archaea, although most homologs show only a low to moderate amino acid identity (30% to 53%) (Fig. 9A; Table S2). They are particularly abundant in the Firmicutes, specifically in the Bacilli class, to which *L. monocytogenes* belongs. Crystal structures have been determined for two Bacilli members: *Alkalihalobacillus halodurans* (PDB, 3E8X; identity, 50%; E value, $6E^{-55}$) and *Lactococcus lactis* (PDB, 3DQP; identity, 31%; E value, $3E^{-19}$). Both are members of the NADP(H)-dependent SDRs, adopting the conserved SDR-typical Rossmann-fold architecture, featuring a parallel seven-stranded $\beta$-sheet flanked on both sides with three $\alpha$-helices (46). A pattern analysis (47) further classified these three YhfK proteins into the atypical SDR subgroup 5 (SDR_a5), which is hallmarked by a glycine-rich NAD(P)-binding motif (Gly-X-X-Gly-X-X-Gly) and a putative active site that includes a highly conserved Tyr-X-X-X-Lys motif, an upstream Ser residue, and a highly conserved Asp residue (46). A sequence alignment showed the conservation of these features in YhfK homologs of several Firmicutes and even of Proteobacteria and Archaea (Fig. 8B). The cellular functions of most proteins in the SDR_a5 subgroup remain unexplored, except for the anaerobilin reductase ChuY of *E. coli* (48, 49), the hydroxycinnamic acid reductase Par1 of *Furfurilactobacillus* spp (50), and the divinyl chlorophyllide 8-vinyl-reductases in some plants and bacteria (51, 52). ChuY is a NADPH-dependent reductase that catalyzes the reduction of anaerobilin, which is a linear tetrapyrrole that is produced by the ChuW-mediated breakdown of heme (49). Thus, it would be interesting to see whether YhfK in bacteria, including *L. monocytogenes*, also participates in heme homeostasis. As in many Gram-positive bacteria, an IsdG-type protein (Lmo2213) was previously reported to degrade heme in *L. monocytogenes* (53). The ability to reduce hydroxycinnamic acids is widespread in heterofermentative lactobacilli, such as *Furfuribacillus* spp., probably because it diminishes their antimicrobial potency and because it allows for the conservation of extra energy from the fermentation of hexoses via the phosphoketolase pathway. Particularly in *F. milii*, the hydroxycinnamic acid reductase Par1 was shown to reduce the ene group of hydroxycinnamic acids and to contribute to fitness during the fermentation of sorghum (54). By analogy, the ability of *L. monocytogenes* to degrade $\alpha,\beta$-unsaturated aldehydes may be an adaptation to grow on plant leaves, which are one of the various environments in which this organism thrives (55).

YhfK$_{L\ mono}$ possesses the highly conserved active site that is typical of the majority of SDRs, with a triad of Ser110-Tyr126-Lys130 residues (numbering based on that of YhfK$_{L.mono}$) that is critical for the protonation and hydride transfer processes (46, 56) (Fig. 8B). Tyr serves as the critical catalytic residue, forming a hydrogen bond with both the substrate carbonyl-oxygen and the nicotinamide ribose to provide/accept a hydride, whereas the adjacent Lys stabilizes the cofactor nicotinamide ribose through hydrogen bonds and lowers the Tyr hydroxyl pKa, together with positively charged nicotinamide, to promote the proton transfer. The Ser residue stabilizes the proton delivery network by forming a hydrogen bond with the carbonyl oxygen of the substrate. Besides the catalytic site, YhfK also contains several residues that are predicted to constitute an NADP(H)-binding domain, including a conserved Gly-rich segment (Gly7 to Gly12) (participates in interactions with the adenine ribose and the central diphosphate group of NADP+), Arg33 and Ala51-Leu53 (adenine-binding moiety), Thr70-Ser73 (ribose-binding moiety), and Pro149-Leu152 (nicotinamide-binding moiety). Interestingly, Leu53 was found to be replaced by Phe in one of our t-CIN tolerant mutants, which might have an effect on the coenzyme binding. The residues of the active site and the NAD(P)H binding site were not affected in the other YhfK variants.

**Conclusions.** By isolating and characterizing the t-CIN tolerant mutants, we were able to identify the SDR superfamily enzyme YhfK of *L. monocytogenes* to be an "ene reductase" that catalyzed the reduction of the C=C bond of $\alpha,\beta$-unsaturated aldehydes. To

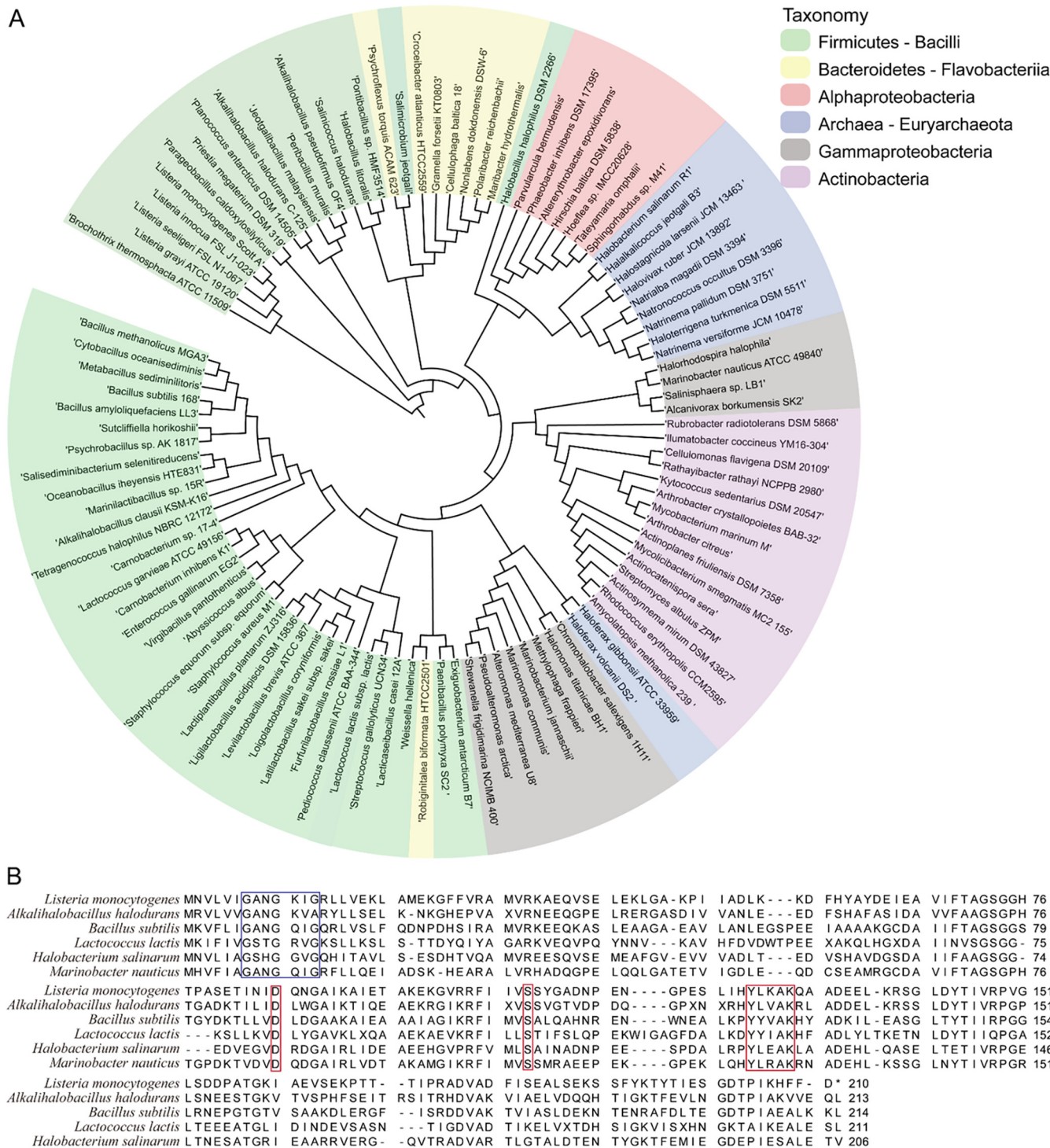

**FIG 8** (A) Phylogenetic analysis of YhfK. The phylogenetic tree was constructed with a set of YhfK homologs from bacterial and archaeal origin using the neighbor-joining approach with 300 bootstrap replicates. The taxonomy of organisms is indicated with colors. (B) Amino acid sequence alignment of YhfK homologs from *Alkalihalobacillus halodurans*, *Lactococcus lactis*, *L. monocytogenes*, and *B. subtilis* (Firmicutes), *Marinobacter nauticus* (Proteobacteria), and *Halobacterium salinarum* (Archaea), using the MUSCLE alignment program (59). The position of the conserved atypical SDR domains (46), including a Gly-rich dinucleotide binding site motif (Gly-X-X-Gly-X-X-Gly) (blue box) and the active site motif, including the Tyr-X-X-X-Lys motif as well as the upstream Ser and Asp residues (red box), all of which are marked. The accession numbers of the amino acid sequences are listed in Table S2.

**TABLE 1** Bacterial strains and plasmids used in this work

| Species | Designation in this work | Description | Reference |
|---|---|---|---|
| *L. monocytogenes* | WT | wild-type strain Scott A, obtained from ILSI strain collection | 60 |
| | WT/pIMK2 | WT with pIMK2 integrated, Km$^R$ | This work |
| | $\Delta yhfK$ | WT with in-frame deletion of *yhfK* | This work |
| | $\Delta yhfK$/pIMK2 | $\Delta yhfK$ with pIMK2 integrated, Km$^R$ | This work |
| | $\Delta yhfK$/pIMK2::*yhfK* | $\Delta yhfK$ with pIMK2::*yhfK* integrated, Km$^R$ | This work |
| | $\Delta yhfK$/pIMK2::*yhfK*(M1) | $\Delta yhfK$ with pIMK2::*yhfK* (M1) integrated, Km$^R$ | This work |
| | $\Delta yhfK$/pIMK2::*yhfK* (M2) | $\Delta yhfK$ with pIMK2::*yhfK* (M2) integrated, Km$^R$ | This work |
| | $\Delta yhfK$/pIMK2::*yhfK* (M3) | $\Delta yhfK$ with pIMK2::*yhfK* (M3) integrated, Km$^R$ | This work |
| | $\Delta yhfK$/pIMK2::*yhfK* (M4) | $\Delta yhfK$ with pIMK2::*yhfK* (M4) integrated, Km$^R$ | This work |
| *E. coli* | S17-1 $\lambda$pir | Donor for plasmid conjugation | 61 |
| | DH5-$\alpha$ | Host strain for plasmid constructs | 62 |
| *B. subtilis* | WT | Strain 168 | BGSC$^a$ |
| | BKK 10260 or $\Delta yhfK$::*km* | WT with *yhfK* replaced by a kanamycin resistance cassette, Km$^R$ | 63 |
| | BKE 10260 or $\Delta yhfK$::*erm* | WT with *yhfK* replaced by an erythromycin resistance cassette, Em$^R$ | 63 |

| Plasmid | Description | | Reference |
|---|---|---|---|
| pIMK2 | Site-specific listerial integrative vector, Phelp promoter for constitutive overexpression, 6.2 kb, Km$^R$ | | 29 |
| pIMK2::*yhfK* | pIMK2 with *yhfK* gene from Scott A under control of pHelp promotor | | This work |
| pHoss1 | Plasmid for gene deletion in *L. monocytogenes*, negative selection based on *secY* antisense RNA, 8,995 bp, Amp$^R$, Ery$^R$ | | 57 |
| pHoss::*yhfK*-LR | pHoss1 with 1 kb flanking fragments upstream and downstream of *yhfK* inserted | | This work |

$^a$Bacillus Genetic Stock Centre (http://www.bgsc.org/).

our knowledge, this is the first bacterial SDR enzyme exhibiting this specificity, and these enzymes may be of interest for biocatalytic applications. The function of YhfK and its catalytic mechanism can be further investigated by enzyme kinetic studies utilizing purified recombinant YhfK and by an analysis of crystal structures of YhfK in complexes with and without cofactors and substrates. The physiological or ecological function of YhfK in *L. monocytogenes* remains an open question. Our work showed the enzyme to be able to detoxify reactive $\alpha,\beta$-unsaturated aldehydes that are generated not only by the host during the infection process (4-HNE), but also by plants on which the bacteria may reside outside the host (t-CIN, t-HEX, citral). However, a role in a metabolic pathway, such as heme degradation, also cannot be excluded. The wide distribution and conservation of YhfK-like enzymes in bacteria suggests an important role and warrants further investigation.

## MATERIALS AND METHODS

**Bacterial strains, plasmids, and growth conditions.** The bacterial strains and plasmids used in this work are listed in Table 1. The *L. monocytogenes* strains were grown at 30°C in brain heart infusion (BHI, Oxoid, Hampshire, United Kingdom) medium, whereas the *E. coli* and *B. subtilis* strains were grown in Luria-Bertani (LB; 10 g/L tryptone, 5 g/L yeast extract, 5 g/L NaCl) medium at 37°C. Agar was added at 1.5% for solid media. The media were supplemented with kanamycin (Km; AppliChem, Darmstadt, Germany), erythromycin (Em; Acros Organics, NJ, USA), ampicillin (Amp; Thermo Fisher Scientific), and anhydrotetracycline (ATc; CaymanChem, MI, USA) when appropriate. The antimicrobial essential oil compounds used in this work include t-CIN (Acros Organics), *trans*-2-hexenal (Sigma-Aldrich, Saint Louis, MO, USA), and 4-hydroxy-2-nonenal (Abcam, Cambridge, GB).

**The isolation of t-CIN tolerant strains via experimental evolution.** The experimental evolution of WT *L. monocytogenes* for increased t-CIN tolerance was performed using six independent parallel lineages, according to the selection scheme shown in Fig. 9. Briefly, 6 colonies were grown overnight in 4 mL BHI broth with shaking, and they were then diluted 1,000-fold in fresh BHI broth with 3 or 4 mM t-CIN. An additional culture with only the equivalent amount of the solvent (ethanol without t-CIN) was included as a control without selection pressure. Two hundred $\mu$L portions of the diluted cultures were transferred into a 96-well microplate, covered with a foil (Greiner Bio-One EASYseal Adhesive Microplate Sealer, Thermo Fisher Scientific), and incubated at 30°C with continuous shaking (250 rpm). When the turbidity reached the level of a stationary-phase culture, the cultures were again diluted 1:1,000 in fresh medium in a new microplate for another round of growth, and this cycle was repeated nine times with a t-CIN concentration of 3 mM or seven times at a t-CIN concentration of 4 mM. After each round, a portion of the cultures was diluted $10^5$-fold, and 100 $\mu$L were spread on BHI agar. A 6 mm sterile Whatman filter paper disc that had been impregnated with 10 $\mu$L pure t-CIN was then placed in the center of the agar plates. After incubation at 30°C for 2 days, 16 colonies that had formed near the edge of the inhibition halo were picked and streaked on BHI agar. The resistance of one colony from each of the 16 isolates against t-CIN was evaluated via a microplate growth assay (see below). The evolution experiment

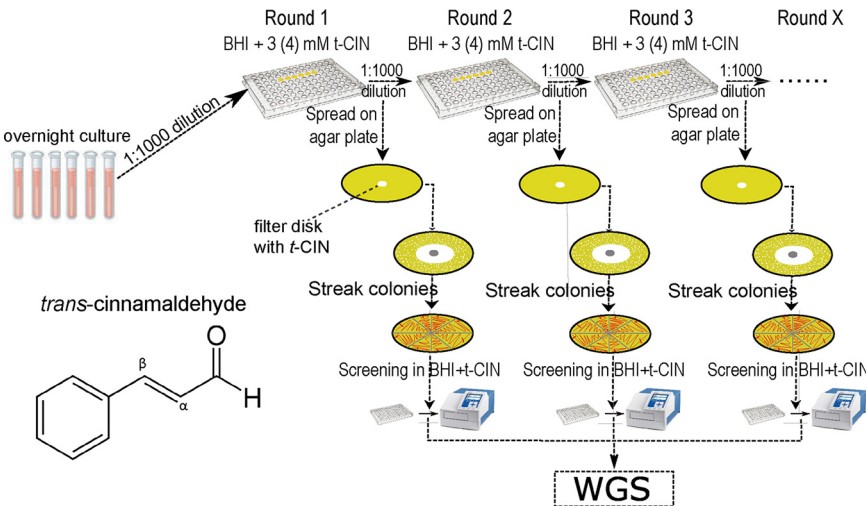

**FIG 9** Selection scheme of the experimental evolution for increased t-CIN tolerance in *L. monocytogenes* Scott A. Isolates with enhanced t-CIN tolerance emerged in seven independent lineages with t-CIN supplementation after nine (in 3 mM t-CIN) or seven (in 4 mM t-CIN) rounds of subculture, but they did not emerge in the control cultures without t-CIN.

was continued until isolates with enhanced t-CIN tolerance emerged. A selection of these isolates from independent cultures was sent for whole-genome sequencing to analyze the presence of mutations.

**Whole-genome sequencing.** Genomic DNA was extracted from overnight cultures in BHI at 30°C with a GeneJET Genomic DNA Purification Kit (Thermo Fisher Scientific). The quality and concentration of the DNA were quantified via gel electrophoresis as well as spectrophotometric NanoDrop and Qubit (Thermo Fisher Scientific) analyses. Paired-end libraries were constructed with a NEBNext Ultra DNA Library Prep Kit (NEB, Ipswich, MA, USA) and were sequenced at VIB Nucleomics Core (Leuven, Belgium) with an Illumina MiSeq sequencer (Illumina, San Diego, CA, USA). The sequence assembly and analysis were done using the CLC Genomic Workbench software package (Qiagen, Hilden, Germany) to identify the mutations that were acquired in the evolved strains, compared to the parental strain, and these mutations were subsequently verified via PCR and Sanger sequencing (Macrogen Europe, Amsterdam, Netherlands).

**Deletion and genetic complementation of *yhfK*.** The pHoss1 plasmid was utilized to generate a markerless, in-frame deletion mutant of *yhfK* (NCBI accession number, CM001159; locus tag, LMOSA_3600) in *L. monocytogenes* Scott A, as previously described (57). First, approximately 1 kb fragments immediately upstream and downstream of *yhfK* were amplified separately with the yhfK-KO-A/B and yhfK-KO-C/D primer pairs, respectively (Table 2). The purified products were joined via overlap extension PCR with the primers yhfK-KO-A and yhfK-KO-D, and the resulting approximately 2 kb fragment and pHoss1 were then digested with BspHI/PstI and NcoI/PstI restriction enzymes, respectively, ligated, and transformed into *E. coli* DH5α. Then, the construct was verified via PCR and Sanger sequencing with the pHO-CK-F/pHO-CK-R primers. The plasmid was subsequently electroporated into *L. monocytogenes* Scott A, using the procedure of (29), and purified clones were then restreaked three successive times on BHI agar with 10 μg/mL Em at 42°C to enforce plasmid integration, and passed two times overnight in BHI broth without Em at 30°C. The reversal of plasmid integration was then enforced by spreading a diluted culture on BHI agar plates with 2 μg/mL ATc to induce the *secY* antisense RNA counterselection system of pHoss1. Clones in which *yhfK* was successfully deleted were identified via colony PCR with the yhfK-KO-A and yhfK-KO-D primers and Sanger sequencing.

**TABLE 2** Primers used in this work

| Primer | Sequence (5′ – 3′)ᵃ | Reference |
|---|---|---|
| yhfK_BspHI | ATATAT<u>TCATGA</u>ATGTACTCGTAATTGGCGCAAA | This work |
| yhfK_SalI | ATATAT<u>GTCGAC</u>ATGAGCGCGTAATTTGGCTCAT | This work |
| yhfK-KO-A | ATAT<u>GGTACC</u>CATAGGGAAATTACGAACTAG | This work |
| yhfK-KO-B | ATTCATTGCTAATCTCCTCCTA | This work |
| yhfK-KO-C | TAGGAGGAGATTAGCAATGAATGACACACCCATTAAACATTT | This work |
| yhfK-KO-D | ATAT<u>CTGCAG</u>CTGGTAAGGTTGAAAGACAA | This work |
| pIMK_REV | CCTATCACCTCAAATGGTTCG | 7 |
| pIMK_FW | GAGTCAGTGAGCGAGGAAGC | 7 |
| NC16(II) | GTCAAAACATACGCTCTTATCGATTC | This work |
| pHO-CK-F | ACGATTGATGCAGTGATGTAGG | This work |
| pHO-CK-R | CGGTCCAATGATCGAAGTTAGG | This work |

ᵃRestriction sites are underlined: NcoI (CCATGG), SalI (GTCGAC), BspHI (TCATGA), PstI (CTGCAG), NdeI (CATATG), and XhoI (CTCGAG).

For the genetic complementation of the Δ*yhfK* mutant, the WT and evolved *yhfK* alleles were amplified using the yhfK_BspHI/yhfK_SalI primer pair (Table 2). The products were cleaved with BspHI and SalI and cloned into the integrational pIMK2 plasmid, which was opened with NcoI and SalI and was introduced in the *L. monocytogenes* Δ*yhfK* mutant via conjugation from *E. coli* S17-1 λpir. Successful integration was confirmed via PCR with the primer yhfK_BspHI, NC16(II) (Table 2), which anneals near and points toward the chromosomal plasmid integration site, and Sanger sequencing with the primers pIMK_FW and pIMK_REV, which point toward the *yhfK* gene from both sides of the pIMK2 cloning site. The complemented strain was designated Δ*yhfK/pIMK2::yhfK*. The WT and Δ*yhfK L. monocytogenes* strains carrying an empty integrated pIMK2 plasmid were constructed in a similar way to serve as controls.

**Growth assays.** Growth curves were established with an automated microplate reader (Multiskan Ascent or Multiskan FC, Thermo Fisher Scientific). Briefly, overnight cultures were adjusted to the same optical density (OD, 600 nm) in each experiment and diluted 1,000-fold in appropriately supplemented BHI broth. 200 μL aliquots were transferred into a 96-well microplate, which was sealed with a cover foil and incubated at 30°C in an automated microplate reader with shaking at 960 rpm. The OD value was read every 15 or 30 min. The Excel add-in package DMFit (Quadram Institute Bioscience, Norwich, United Kingdom) was used to determine the maximum growth rate ($\mu_{max}$), the lag phase time ($\lambda$), and the maximal OD ($OD_{max}$) value at the stationary phase through the Baranyi and Roberts model (58).

**Analysis of t-CIN and its metabolites.** Headspace solid-phase microextraction gas chromatography mass spectrometry (HS-SPME-GC-MS) analysis was used to monitor the concentrations of t-CIN and its metabolites in bacterial cultures, following a previously described procedure, with minor modifications (7). First, overnight cultures adjusted to the same OD were 1,000-fold diluted in 70 mL fresh BHI broth with 1 mM t-CIN in 250 mL glass flasks with the caps tightly screwed so as to avoid the evaporation of the compounds. An uninoculated flask was included as a control. The flasks were incubated at 30°C with continuous shaking (250 rpm), and 2 mL samples were taken every 3 h. A small portion was used to determine a plate count, and the supernatant was obtained after clearing the remainder of the sample via centrifugation (4,000 × g, 5 min), which was flash frozen with liquid nitrogen and stored at −80°C. Immediately before the GC-MS analysis, 1 mL saturated NaCl solution, 100 μL 0.01 mM ethyl benzoate (internal standard) (≥99%, Sigma-Aldrich), and 1 mL unfrozen sample were pipetted into a 10 mL amber glass vial with a screw-cap that had a PTFE/silicon septum seal and was additionally sealed with parafilm. The presence of t-CIN, its degradation products (3-phenyl-2-propenol, 3-phenylpropanal, and 3-phenylpropanol), and ethyl benzoate in the mass spectrum were identified using the NIST 14 Mass Spectral Library and the NIST Mass Spectral Search Program Version 2.2 (National Institute of Standards and Technology, Gaithersburg, MD, USA), and the ions that were monitored for each compound are listed in Table S1. For every strain, three independent cultures were measured.

**Survival assay of *L. monocytogenes* in the presence of 4-hydroxy-2-nonenal (4-HNE).** Overnight cultures were diluted to an $OD_{600}$ of approximately 1, washed twice, resuspended, and diluted 1:1,000 in 10 mM potassium phosphate buffer (PPB) (pH 7.0) in PCR tubes. 4-HNE solution (10 mM in PPB) was added to a final concentration of 2 mM. A control cell suspension, to which the equivalent amount of sterile PPB (without 4-HNE) was added, was included. After incubation at 30°C for 6 h, the cell suspensions were serially diluted and plated on BHI plates to count the survivors.

**Phylogenetic analysis of YhfK homologs.** The NCBI (National Centre for Biotechnology Information) BLASTP algorithm was used to search for homologs of YhfK from *L. monocytogenes* Scott A. The prokaryotic sequences that are listed in the KEGG (Kyoto Encyclopedia of Genes and Genomes) Organisms database with an amino acid sequence identity of ≥30% over at least 90% of the entire sequence and an E value of ≤1E$^{-10}$ were retrieved and aligned using MUSCLE (59) as well as visualized using CLC Genomic Workbench (Qiagen). A phylogenic tree was then constructed in CLC Genomic Workbench using the neighbor-joining approach with 300 bootstrap replicates. For visualization, the phylogeny tree only displays 104 representative sequences (Table S2) with high identity values from different genera and species. Finally, the constructed phylogenetic tree was edited and annotated using the Interactive Tree of Life web server tool (https://itol.embl.de/).

**Statistical analysis.** Data from the growth assay and GC-MS experiment are presented as means ± standard deviation (SD) from three independent cultures (biological replicates) for each strain. The significance of each mean difference was calculated using Tukey's honestly significant difference (Tukey's HSD) test for the growth parameters ($\lambda$, $\mu$max, and $OD_{max}$) or Student's *t* test (two-tailed) for the survival assay in the presence of 4-HNE using GraphPad PRISM 7.0 (GraphPad, San Diego, CA, USA). A *P* value of <0.05 was considered to be indicative of a statistically significant result.

**Data availability.** All data are included in the article and supplemental material.

## SUPPLEMENTAL MATERIAL

Supplemental material is available online only.

**SUPPLEMENTAL FILE 1**, PDF file, 0.5 MB.

## ACKNOWLEDGMENTS

This work was supported by research grants from the Research Foundation-Flanders (FWO) (G.0C77.14N) and from the KU Leuven Research Fund (METH/14/03).

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
