## [Reviewer comments · Microbiology Spectrum]

Microbiology Spectrum

Experimental Evolution Reveals a Novel Ene Reductase that Detoxifies α,β -Unsaturated Aldehydes in *Listeria monocytogenes*

Sun Lei, Buvé Carolien, Van Loey Ann, and Chris Michiels

Corresponding Author(s): Chris Michiels, Katholieke Universiteit Leuven

Review Timeline:

Submission Date:	November 30, 2022
Editorial Decision:	March 6, 2023
Revision Received:	March 15, 2023
Accepted:	March 17, 2023

Editor: Cezar Khursigara

Reviewer(s): Disclosure of reviewer identity is with reference to reviewer comments included in decision letter(s). The following individuals involved in review of your submission have agreed to reveal their identity: Michael G. Ganzle (Reviewer #1)

Transaction Report:

DOI: <https://doi.org/10.1128/spectrum.04877-22>

March 6, 2023

Prof. Chris W. Michiels
Katholieke Universiteit Leuven
Laboratory of Food Microbiology and Leuven Food Science and Nutrition Research Centre (LFoRCe)
Kasteelpark Arenberg 22 - box 2457
Heverlee 3001
Belgium

Re: Spectrum04877-22 (Experimental Evolution Reveals a Novel Ene Reductase that Detoxifies α,β -Unsaturated Aldehydes in *Listeria monocytogenes*)

Dear Prof. Chris W. Michiels:

One expert has reviewed your manuscript and we agree that it is great work. The reviewer highlights some minor changes that should be considered upon resubmission. I apologize for the delay in getting this back to you.

Thank you for submitting your manuscript to Microbiology Spectrum. As you will see your paper is very close to acceptance. Please modify the manuscript along the lines I have recommended. As these revisions are quite minor, I expect that you should be able to turn in the revised paper in less than 30 days, if not sooner. If your manuscript was reviewed, you will find the reviewers' comments below.

When submitting the revised version of your paper, please provide (1) point-by-point responses to the issues raised by the reviewers as file type "Response to Reviewers," not in your cover letter, and (2) a PDF file that indicates the changes from the original submission (by highlighting or underlining the changes) as file type "Marked Up Manuscript - For Review Only". Please use this link to submit your revised manuscript. Detailed instructions on submitting your revised paper are below.

Link Not Available

Sincerely,

Cezar Khursigara

Reviewer comments:

Reviewer #1 (Comments for the Author):

The manuscript describes the role of YhfK in the resistance of *Listeria monocytogenes* to cinnamaldehyde and other beta-unsaturated aldehydes. The manuscript uses an elegant experimental design that includes experimental evolution, the comprehensive characterization of (complemented) YhfK k.o. mutants, and analytical determination of substrates and products of YhfK. Overall, the manuscript makes an important contribution to our understanding on how environmental and plant-associated microbes defend themselves against plant secondary metabolites with antimicrobial activity.

Minor comments for improvement of the manuscript are listed below. The only somewhat major comment relates to the role of YhfK for *L. monocytogenes*: If we take an ecological / evolutionary approach to the topics, then this is more likely a story that informs on resistance of environmental and plant-associated microbes against antimicrobial plant secondary metabolites.

Specific comments:

Line 41. Nitrite does not inhibit *Listeria monocytogenes* and benzoate and sorbate are not under scrutiny for adverse health effects. The use of nitrites continues owing to its specific activity against *C. botulinum*; replacement of other preservatives aims to produce minimally processed foods or foods with a "kitchen friendly" ingredient list.

line 45. Please write trans-cinnamaldehyde in full on first use.

line 53. trans-2-hexenal has an odor threshold of 2.5 $\mu\text{g} / \text{L}$ but the MIC is more than 1,000 fold higher - hexenal is an unlikely

candidate as a food preservative. Hexenal is produced when grass or other leafy plants are cut or injured and may be relevant in environmental habitats of *Listeria monocytogenes*.

Figure 2 panel B. Depending on whether the authors wish to emphasize lag phase and growth rate, or the final optical density, the growth curves are preferably shown with log-transformed values for bacterial biomass. Because values for lag phase, growth rate and final population density are shown in panel C, the graph can be omitted altogether.

Figure 3 - as Figure 2.

Lines 297 to 321. YhfK is also 38% identical to NADH-dependent reductases in *Furfurilactobacillus mii* - and Par1 are active on hydroxycinnamic acids but Par2 has an unknown substrate specificity. It seems enzymes that accept carboxylic acids adjacent to the double bond are closely related to YhfK.

line 338 and 409. It is extremely unlikely that active (millimolar) concentrations of hexenal are formed in vivo - inflamed tissue usually does not smell like green grass. In addition, *Listeria* is an accidental pathogen but has not evolved to infect humans or animals. It may be more likely that hexenal and related aldehydes are formed in the establishment niche of *Listeria monocytogenes*, environmental habitats, when leafy plants are cut or injured.

lines 341 to 352. See comment to hydroxycinnamic acid reductases above.

line 412. The introduction starts with food safety and food preservation - it may be worthwhile to pick this up at the end of the discussion section?

Preparing Revision Guidelines

Please return the manuscript within 60 days; if you cannot complete the modification within this time period, please contact me. If you do not wish to modify the manuscript and prefer to submit it to another journal, please notify me of your decision immediately so that the manuscript may be formally withdrawn from consideration by Microbiology Spectrum.

Spectrum04877R1: Response to reviewers

Note: line numbers referring to changes correspond to the marked manuscript

Reviewer #1 (Comments for the Author):

The manuscript describes the role of YhfK in the resistance of *Listeria monocytogenes* to cinnamaldehyde and other beta-unsaturated aldehydes. The manuscript uses an elegant experimental design that includes experimental evolution, the comprehensive characterization of (complemented) YhfK k.o. mutants, and analytical determination of substrates and products of YhfK. Overall, the manuscript makes an important contribution to our understanding on how environmental and plant-associated microbes defend themselves against plant secondary metabolites with antimicrobial activity. Minor comments for improvement of the manuscript are listed below. The only somewhat major comments relates to the role of YhfK for *L. monocytogenes*: If we take an ecological / evolutionary approach to the topics, then this is more likely a story that informs on resistance of environmental and plant-associated microbes against antimicrobial plant secondary metabolites.

We thank the reviewer for the appreciation for our work.

Specific comments:

Line 41. Nitrite does not inhibit *Listeria monocytogenes* and benzoate and sorbate are not under scrutiny for adverse health effects. The use of nitrites continues owing to its specific activity against *C. botulinum*; replacement of other preservatives aims to produce minimally processed foods or foods with a "kitchen friendly" ingredient list.

There is disagreement about the effectiveness of nitrite against *L. monocytogenes*, but several studies indicate that it helps control this pathogen in cured meat products, particularly when used in combination with other hurdles, e.g. ascorbate. (King et al., 2016. Modeling the Impact of Ingoing Sodium Nitrite, Sodium Ascorbate, and Residual Nitrite Concentrations on Growth Parameters of *Listeria monocytogenes* in Cooked, Cured Pork Sausage. *J. Food Prot.* <https://doi.org/10.4315/0362-028X.JFP-15-322>). Further, we do not want to suggest that there are important safety concerns with sorbate and benzoate, but only that the safety of these and other food additives keeps being questioned, and that their safety is regularly reassessed, for example by the European Food Safety Authority (e.g. for sorbate: <https://www.efsa.europa.eu/en/efsajournal/pub/5625>). Therefore, we have maintained the statements, but have replaced one of the references (#5) by a more pertinent one:

Erickson, M.C.; Doyle M.P. The Challenges of Eliminating or Substituting Antimicrobial Preservatives in Foods. *Ann. Rev. Food Sci. Technol.* **2017**, *8*, 371-390.

line 45. Please write trans-cinnamaldehyde in full on first use.

Done

line 53. trans-2-hexenal has an odor threshold of 2.5 µg / L but the MIC is more than 1,000 fold higher - hexenal is an unlikely candidate as a food preservative. Hexenal is produced when grass or other leafy plants are cut or injured and may be relevant in environmental habitats of *Listeria monocytogenes*.

The use of plant essential oil components as food preservatives is indeed often limited by the off-flavors of odors they impart. However, the odor and flavor thresholds depend on the matrix and are typically higher in foods (e.g. for trans-2-hexenal it is 420 – 1125 µg/kg in oil); Furthermore, there are various ways to reduce the sensorial impact and at the same time enhance the antimicrobial activity, such as micro-

encapsulation combining multiple compounds that act synergistically. However, we think it is better not to elaborate on these applied aspects since this is not the focus of our work.

Figure 2 panel B. Depending on whether the authors wish to emphasize lag phase and growth rate, or the final optical density, the growth curves are preferably shown with log-transformed values for bacterial biomass. Because values for lag phase, growth rate and final population density are shown in panel C, the graph can be omitted altogether.

Figure 3 - as Figure 2.

We feel that showing the graphical presentation of the growth curves and the key kinetic parameters (lag phase, exponential growth rate and maximal density) side by side gives the reader a more rapid and complete insight in the strain differences than if only a table with the kinetic parameters would be shown. The transformation of the y-axis to a log scale would be useful when the values span a wide range of several decades, but since the OD range is less than one decade we do not see the added value. It is also quite common to present OD values on a linear scale.

Lines 297 to 321. YhfK is also 38% identical to NADH-dependent reductases in *Furfurilactobacillus mii* - and Par1 are active on hydroxycinnamic acids but Par2 has an unknown substrate specificity. It seems enzymes that accept carboxylic acids adjacent to the double bond are closely related to YhfK.

Although we had identified a *Furfurilactobacillus* homolog (see Table S2; unchanged), we were not aware that its function had been reported. This is an interesting point, and we have included a few sentences about it in the discussion (L382-383 and L388-392).

line 338 and 409. It is extremely unlikely that active (millimolar) concentrations of hexenal are formed in vivo - inflamed tissue usually does not smell like green grass.

The compound discussed in relation to infection is 4-hydroxynonenal (4-HNE), not hexenal (which indeed has a mowed green grass smell). It is correct that 4-HNE is not formed at mM concentrations in vivo, but one has to consider that the presence of other antimicrobial substances generated during the oxidative burst is likely to increase the sensitivity of the pathogen for 4-HNE. Whether or not 4-HNE plays a role in the antibacterial defense during infection remains to be established, but a recent study that we cited provides some evidence supporting such a role. Upon rereading this paper, we noticed that we somewhat overstated the conclusions, and we have attenuated the corresponding sentence in the revised version of our manuscript (L337-340).

In addition, *Listeria* is an accidental pathogens but has not evolved to infect humans or animals.

We disagree with the reviewer on this point. *L. monocytogenes* has a dual lifestyle: it is a successful saprophyte that thrives well in soil, decaying plant material, and water, but it also has highly specialized mechanisms controlled by complex regulatory systems to infect humans and survive intracellularly. See for example: Radoshevich, L., Cossart, P. *Listeria monocytogenes*: towards a complete picture of its physiology and pathogenesis. *Nat Rev Microbiol* 16, 32–46 (2018).
<https://doi.org/10.1038/nrmicro.2017.126>

It may be more likely that hexenal and related aldehydes are form in the establishment niche of *Listeria monocytogenes*, environmental habitats, when leafy plants are cut or injured.

At this point we can only speculate about possible roles of YhfK in different non-host environments and in virulence. Since this role was not addressed in the present paper, we prefer to list some possibilities without speculating too much. We have added a sentence in the discussion to point to the possible role to facilitate growth on plant leaves because this is indeed an interesting possibility (L392-394).

lines 341 to 352. See comment to hydroxycinnamic acid reductases above.

The hydroxycinnamic acid reductases from *Furfurilactobacillus* spp. have been included in the discussion elsewhere (see answer to a previous comment). The paragraph lines 341-352 (original Ms) specifically discusses the “old yellow enzymes”, to which the hydroxycinnamic acid reductases from *Furfurilactobacillus* spp. show no significant sequence similarity (as verified again by us during this revision).

line 412. The introduction starts with food safety and food preservation - it may be worthwhile to pick this up at the end of the discussion section?

The story is indeed introduced in a context of food preservation (possible use of t-cinnamaldehyde as an antimicrobial food preservative), but the research questions and the results are rather more of a basic scientific interest. That is also how the whole story is presented and discussed. We do present any data directly related to the food safety aspect, and we therefore believe it would feel artificial to try to close the discussion by returning to food safety.

Additional correction: reference 59 (original numbering) was omitted since it was not cited in the text.

March 17, 2023

Prof. Chris W. Michiels
Katholieke Universiteit Leuven
Laboratory of Food Microbiology and Leuven Food Science and Nutrition Research Centre (LFoRCe)
Kasteelpark Arenberg 22 - box 2457
Heverlee 3001
Belgium

Re: Spectrum04877-22R1 (Experimental Evolution Reveals a Novel Ene Reductase that Detoxifies α,β -Unsaturated Aldehydes in *Listeria monocytogenes*)

Dear Prof. Chris W. Michiels:

Your manuscript has been accepted, and I am forwarding it to the ASM Journals Department for publication. You will be notified when your proofs are ready to be viewed.

Sincerely,

Cezar Khursigara
Editor, Microbiology Spectrum
